# Manufacturing and Properties of Binary Blend from Bacterial Polyester Poly(3-hydroxybutyrate-*co*-3-hydroxyhexanoate) and Poly(caprolactone) with Improved Toughness

**DOI:** 10.3390/polym12051118

**Published:** 2020-05-14

**Authors:** Juan Ivorra-Martinez, Isabel Verdu, Octavio Fenollar, Lourdes Sanchez-Nacher, Rafael Balart, Luis Quiles-Carrillo

**Affiliations:** Technological Institute of Materials (ITM), Universitat Politècnica de València (UPV), Plaza Ferrándiz y Carbonell 1, 03801 Alcoy, Spain; isvergar@epsa.upv.es (I.V.); ocfegi@epsa.upv.es (O.F.); lsanchez@mcm.upv.es (L.S.-N.); rbalart@mcm.upv.es (R.B.); luiquic1@epsa.upv.es (L.Q.-C.)

**Keywords:** bacterial polyesters, poly(3-hydroxybutyrate-*co*-3hydroxyhexanoate)—PHBH, poly(ε-caprolactone)—PCL, binary blends, improved toughness, mechanical and thermal characterization

## Abstract

Polyhydroxyalkanoates (PHAs) represent a promising group of bacterial polyesters for new applications. Poly(3-hydroxybutyrate-*co*-3-hydroxyhexanoate) (PHBH) is a very promising bacterial polyester with potential uses in the packaging industry; nevertheless, as with many (almost all) bacterial polyesters, PHBH undergoes secondary crystallization (aging) which leads to an embrittlement. To overcome or minimize this, in the present work a flexible petroleum-derived polyester, namely poly(ε-caprolactone), was used to obtain PHBH/PCL blends with different compositions (from 0 to 40 PCL wt %) using extrusion followed by injection moulding. The thermal analysis of the binary blends was studied by means of differential scanning calorimetry (DSC) and thermogravimetry (TGA). Both TGA and DSC revealed immiscibility between PHBH and PCL. Mechanical dynamic thermal analysis (DMTA) allowed a precise determination of the glass transition temperatures (*T*_g_) as a function of the blend composition. By means of field emission scanning electron microscopy (FESEM), an internal structure formed by two phases was observed, with a PHBH-rich matrix phase and a finely dispersed PCL-rich phase. These results confirmed the immiscibility between these two biopolymers. However, the mechanical properties obtained through tensile and Charpy tests, indicated that the addition of PCL to PHBH considerably improved toughness. PHBH/PCL blends containing 40 PCL wt % offered an impact resistance double that of neat PHBH. PCL addition also contributed to a decrease in brittleness and an improvement in toughness and some other ductile properties. As expected, an increase in ductile properties resulted in a decrease in some mechanical resistant properties, e.g., the modulus and the strength (in tensile and flexural conditions) decreased with increasing wt % PCL in PHBH/PCL blends.

## 1. Introduction

Nowadays, awareness of environmental protection, sustainable development, and the use of renewable energies has become a priority for our society. The high volume of wastes generated that are harmful for the environment, oceans, ecosystems, and so on, has become a major problem to be solved. Furthermore, the waste generated in a consumer society, such as the present one, comes mainly from the packaging sector. This need has favoured the development of new environmentally friendly materials [1]. For this reason, the use of the so-called biopolymers is increasing in the packaging sector. Most of these materials are obtained from renewable resources and they are, in many cases, biodegradable (or compostable in controlled compost soil). Therefore, they positively contribute to minimizing plastic wastes, thus reducing the carbon footprint and also contributing to circular economies by upgrading industrial wastes [2] and/or by-products [3].

In this area, researchers have successfully developed new polymeric materials from renewable and/or biodegradable sources. These important research works have allowed the optimization of interesting biopolymers to be scaled in the industry such as poly(lactic acid)—PLA [4], poly(hydroxyalkanoates)—PHA [5], thermoplastic starch—TPS [6], poly(ε-caprolactone)—PCL [7], and poly(butylene succinate)—PBS [8], among others. Biopolymers can perfectly replace some petroleum-derived polymers [9], since they offer similar performance to most commodities and some engineering plastic. Biopolyesters are an interesting group which includes petroleum-based polymers such as poly(glycolic acid)—PGA, poly(butylene succinate)—PBS, poly(butylene adipate-*co*-terephthalate)—PBAT, and PCL, among others. However, biopolyesters also include bacterial polyesters (PHAs) and some starch-derived polymers such as PLA. The main advantage of polyesters (from both natural or fossil resources) is that they can undergo biodegradation (disintegration in controlled conditions with special compost soil), through the action of microorganisms. This makes composting an important and simple sustainable option for the management of these wastes [10].

Nowadays, biopolymers produced by bacterial fermentation, such as polyhydroxyalkanoates (PHAs), are becoming very promising as there are more than 300 potential PHAs and copolymers. Despite this wide variety, the most commonly used and commercially available PHAs are poly(3-hydroxybutyrate)—P3HB—and poly(3-hydroxybutyrate-*co*-3-hydroxyvalerate)—PHBV [11,12]. PHAs are biologically synthesized polyesters by controlled fermentation with bacteria, such as *Gram-negative* bacteria (*Azobacter*, *Bacillus* and *Pseudomonas*) and *Gram-positive* bacteria (*Rhodococcus*, *Nocardia* and *Streptomyces*). These bacteria, under food stress, can produce energy reserves as intracellular food in the form of granules. These granules are stored in the form of PHAs [13].

Arrieta et al. [10] reported that these bacteria, under feeding conditions of limited macro-elements (such as phosphorus, nitrogen, trace elements or oxygen) and in the presence of an abundant source of carbon (e.g., glucose or sucrose) and/or lipids (e.g., vegetable oils or glycerin), are capable of accumulating up to 60–80 wt % in the form of PHAs. In this way, they can subsist under conditions of food restriction [14,15]. Similar to plants that store energy in the form starch polymer, some bacteria are able to accumulate energy reserves in the form of PHAs [16].

It should be noted that these bacterial polyesters are high-molecular-weight, semi-crystalline, biocompatible thermoplastic polymers. They have very good biodegradability even under environmental conditions. They tend to exhibit rigid behaviour, due to high crystallinity, low thermal stability, and small temperature windows for conventional processing [14,17].

These limitations have been improved by the bacterial synthesis of different copolymers. In this way, a wide range of physical and thermal properties can be tailored, depending on the chemical structure of the used comonomers. More than 150 types of monomers have been successfully synthesized by selecting different raw or modified bacteria and/or the fermentation conditions [13]. The work of Alata et al. [18] reported the effect of medium-length side groups of 3-hydroxyhexanoate (3-HH) units from 5 to 18 mol % on properties of poly(3-hydroxybutyrate-*co*-3-hydroxyhexanoate)—P(3HB-*co*-3HH) or simply PHBH. They observed a remarkable decrease in crystallinity (*χ*_c_) from 41.6% to 25.1% for copolymers containing 5 mol % and 18 mol % 3-HH, respectively. In addition, due to the reduced crystallinity, secondary crystallization is very low for high 3-HH (above 10 mol %) content in P(3-HB-*co*-3-HH). Other studies have also reported similar results, together with an interesting decrease in the melting temperature of P(3HB-*co*-3HH) [15].

PHBH consists of a random copolymer of 3-HB and 3-HH (see Scheme 1). 3-HH medium-length chains act as short branches of the main 3-HB chains; therefore, stereoregularity is lost, and subsequently, crystallinity is remarkably reduced. Besides this, the presence of aleatory 3-HH chains, broadens the melt peak, but the storage modulus and the overall strength is reduced [17,19]. Nevertheless, PHBH copolymers with low 3-HH content undergo physical aging with time (increase in modulus and strength and reduction of ductile properties such as elongation at break, toughness, and impact strength) [20], which is ascribed to secondary crystallization above the glass transition temperature, *T*_g_ [21]. It is worthy to note that typical values of *T*_g_ for P3HB are −5 to 5 °C, and this interval is remarkably reduced to values as low as −38 °C for medium-to-long alkanoate chains, e.g., the *T*_g_ of poly(3-hydroxyhexanoate) is close to −28 °C [22].

Another key issue in the massive use of PHAs at industrial scale is their “relatively” low cost due to the use of renewable resources such as coconut oil, sugarcane, beet, molasses, other vegetable oils, and, most importantly, carbon-rich industrial wastes such as those obtained from agro-food industry or even sludge coming from sewage treatment plants, by selecting the appropriate bacteria or using bacterial engineering to tailor the desired behaviour of a particular bacteria strain [23,24].

These properties make PHBH an environmentally efficient biopolymer suitable for applications in different packaging applications such as disposable plastic bags, food packaging, catering, agricultural mulch films, and so on [25]. However, as mentioned above, most PHAs undergo secondary crystallization or aging that makes them fragile, thus limiting their possible applications [26]. Xu et al. [27] suggested that one disadvantage of PHBH is that the secondary crystallization process is very slow (for low mol % 3-HH), due to the irregularity of its polymer chain. Large spherulites and secondary crystallization give them poor mechanical properties.

Plasticization of PHAs has been studied with an improvement of ductile properties [28]. Since plasticizers are based on low-molecular-weight compounds, they usually show potential migration problems [29]. An interesting approach to overcome this drawback is blending PHBH with another ductile polymer. However, it is important to bear in mind that the selected polymer for the blend must not compromise biodegradation or disintegration in controlled compost soil. Some researchers have already used poly(ε-caprolactone)—PCL—in blends with PHAs. PCL is a semi-crystalline biodegradable polyester with a very low *T*_g_ of about −60 °C, which gives an overall ductile behaviour with high elongation at break [26,30]. The addition of PCL decreases the fragility of the PHAs, reducing the elastic modulus and improving the blend processability. However, its low melting temperature (around 50–60 °C) means the obtained blends should not be used at temperatures above 50–60 °C since dimensional stability could be compromised. Garcia-Garcia et al. [31] reported a noteworthy improvement in the impact behaviour of P3HB by blending it with PCL. In addition, P3HB/PCL blends improved the flexibility and ductility.

The aim of this work is to overcome the intrinsic fragility of a bacterial copolyester, namely poly(3-hydroxybutyrate-*co*-3-hydroxyhexanoate)—PHBH—, by blending with a flexible polyester, namely poly(ε-caprolactone)—PCL. The effect of the incorporation of different amounts of PCL is evaluated by means of mechanical, thermal, thermo-mechanical, and morphological characterization. The evaluation of the results allows the optimum PHBH/PCL blends to be established for applications in the packaging sector that do not compromise the environment at the end-of-life cycle. In this way, it contributes to the reduction of the current serious problem of eliminating the large volume of plastic waste generated by the packaging sector. In addition, these developed formulations could be used in medical applications, as improved toughness is expected with PCL addition and both are resorbable biopolyesters.

## 2. Materials and Methods

### 2.1. Materials

Poly(3-hydroxybutyrate-*co*-3-hydroxyhexanoate)—PHBH—commercial grade ErcrosBio^®^ PH 110 was supplied in pellet form by Ercros S.A. (Barcelona, Spain). This has a density of 1.2 g cm^−3^ and a melt flow index of 1.0 g/10 min, measured at 160 °C. As indicated by the supplier, it is suitable for injection moulded parts, and it can be melt-blended with other polyesters to obtain tailored properties. Regarding poly(ε-caprolactone)—PCL, commercial grade Capa^TM^ 6800, in pellet form, with a mean molecular weight of 80,000 Da, was supplied by Perstorp (Cheshire, UK). This PCL grade has an MFI (Melt Flow Index) of 3 g/10 min at 160 °C.

### 2.2. Manufacturing of PHBH/PCL Binary Blends

PHBH pellets were dried for 8 h at 80 °C, while PCL pellets were dried at 45 °C for 24 h, in an air-circulating oven CARBOLITE Eurotherm 2416 CG (Hope Valley, UK). As has been reported in other works, the typical weight content (wt %) of flexible polymer blended with brittle polymers to improve toughness is comprised in the 20–40 wt % range. In this work, we selected a maximum PCL content of 40 wt %, since at this composition, PCL is still the minor component in the blend [32,33,34]. Garcia et al. [31] studied the whole PHB/PCL system and revealed, as expected, that with above 50 wt % PCL, it is PCL which defines the properties of the blend. Ferry et al. [35] also confirmed a maximum loading of 30 wt % PCL to improve the high brittleness of neat PLA. Then, different wt % of PHBH and PCL (see Table 1) were mechanically mixed in a zipper bag to provide initial homogenization. After that, all compositions were extruded using a twin-screw corotating extruder manufactured by DUPRA S.L. (Alicante, Spain) with a temperature profile (four barrels, from the hopper to the extrusion die) of 110, 120, 130, and 140 °C respectively and a screw speed of 20 rpm. The extruded material was cooled in air and subsequently pelletized for further processing. After pelletizing, the different blends were subjected to injection moulding in a Meteor 270/75 from Mateu & Solé (Barcelona, Spain). The temperature profile in the injection moulding process was 150 °C (hopper), 140, 130, and 120 °C (nozzle) in a heated mould at 60 °C as recommended by the supplier, since this PHBH has a very low melt strength. The filling time was set to 3 s while the cooling time was 60 s. Standard samples (rectangular and dog-bone shape) were obtained for further characterization. After processing, the specimens were stored at room temperature in a vacuum desiccator for 15 days before characterization, due to the secondary crystallization process or aging that PHBH undergoes with time at 25 °C [21,26,27].

### 2.3. Characterizations Techniques

#### 2.3.1. Thermal and Thermomechanical Characterization

The thermal transitions of PHBH/PCL binary blends were analyzed using differential scanning calorimetry (DSC) in a Q2000 DSC from TA Instruments (New Castle, DE, USA). The temperature program was scheduled in three different stages: 1st heating, 1st cooling, and 2nd heating. The first heating was scheduled from −50 to 200 °C. The second stage consisted of a cooling program from 200 °C down to −50 °C (this stage is interesting to remove the thermal history and allow crystallization); finally, a 2nd heating cycle identical to the first one (−50 to 200 °C) was launched. The heating/cooling rates were all set to 10 °C·min^−1^. All the DSC runs were performed in an inert nitrogen atmosphere with a flow rate of 66 mL·min^−1^. In addition to parameters such as the melt peak temperature (*T*_m_), the cold crystallization peak temperatures (*T*_cc_), enthalpies related to the melting (Δ*H*_m_), and cold crystallization (Δ*H*_cc_) processes, the degree of crystallinity, *χ*_c_ (%), was calculated for each polymer in the blend as
(1)$$\chi_{c}\left(\%\right)=\left[\frac{\Delta H_{m}-\Delta H_{\mathrm{cc}}}{\Delta H_{m}^{0}\cdot \left(1-w\right)}\right]\cdot 100$$

The normalized enthalpy values ($\Delta H_{m}^{0}$) for a theoretical 100% crystalline PHBH and PCL were taken as 146 and 156 J·g^−1^, respectively, as reported in the literature [19,36,37]. Finally, the term (1 − w) stands for the actual weight of the polymer whose crystallinity is being evaluated.

To study the thermal degradation, thermogravimetry (TGA) was carried out in a Mettler-Toledo Inc. TGA 851-E thermobalance (Schwerzenbach, Switzerland). The thermal program used in this case was a unique dynamic ramp from 30 °C up to 700 °C at 20 °C·min^−1^ in an N_2_ inert atmosphere with a flow rate of 66 mL·min^−1^. In this analysis, the onset degradation temperature was taken as the temperature related to a mass loss of 2 wt % and was denoted as *T*_2%_.

Dynamic-mechanical thermal analysis, or DMTA, was carried out in a Mettler-Toledo dynamic analyzer DMA1 (Columbus, OH, USA) in single cantilever mode on rectangular samples sized 40 × 10 × 4 mm^3^. Samples were subjected to slightly different temperature ramps since PCL melts at 58–60 °C. The maximum dynamic deflection was 10 μm and the frequency for the sinusoidal stress wave was set to 1 Hz. Thus, for neat PCL the heating range was from −70 to 50 °C to avoid melting. In the case of neat PHBH, the heating ramp was set from −70 to 100 °C, and finally, PHBH/PCL blends were subjected to a heating program from −70 to 70 °C. The heating rate was the same, 2 °C·min^−1^, for all the different scheduled temperature programs

To evaluate the dimensional stability, neat PHBH and PCL, as well as PHBH/PCL blends, were tested in a thermomechanical analyzer (TMA) Q400 from TA Instruments (New Castle, DE, USA) on rectangular samples sized 10 × 10 × 4 mm^3^. The temperature sweep was from −70 to 70 °C, with a constant heating rate of 2 °C min^−1^ and a constant load of 20 mN. The coefficient of linear thermal expansion (CLTE) of all specimens was determined as the slope for the linear correlation between the expansion and temperature, both below and above *T*_g_.

#### 2.3.2. Mechanical Properties

The mechanical characterization of the PHBH/PCL blends was studied by means of tensile, flexural, impact, and hardness tests on five standardized specimens for each test. The tensile and flexure tests were carried out according to ISO 527 and ISO 178 respectively, in an ELIB 30 universal machine from S.A.E. Ibertest (Madrid, Spain). The load cell was 5 kN for both tests. The crosshead rate was 5 mm·min^−1^ for flexural tests and 20 mm·min^−1^ for the tensile tests. 

The impact resistance (absorbed-energy during impact conditions, per unit area) was determined according to ISO 179, using a Charpy pendulum from Metrotec S.A. (San Sebastian, Spain) with an energy of 1-J. A standardized “V-type” notch was produced on standard rectangular samples. 

The hardness properties were measured according to ISO 868. The equipment used was a Shore D hardness tester model 673-D from J. Bot, S.A. (Barcelona, Spain). 

#### 2.3.3. Morphology Characterization

The surface analysis of the fractured specimens from impact tests was performed with a field emission scanning electron microscope (FESEM) model ZEISS ULTRA55 (Oxford Instruments, Abingdon, UK). The working accelerating voltage was 2 kV. Prior to this analysis, the samples were metallized with a gold-palladium alloy in an EMITECH mod. SC7620 sputter coater from Quorum Technologies Ltd. (East Sussex, UK). In a second analysis, the samples were subjected to a selective PCL extraction in acetone at room temperature for 24 h. In this way, PCL can be extracted, and therefore, it is possible to observe more accurately the phase distribution in the developed binary blends [38].

## 3. Results and Discussion

### 3.1. Thermal Properties of PHBH/PCL Blends

Thermal analysis, using DSC of PHBH/PCL binary blends with different amounts of PCL (wt %), and neat PHBH and PCL, was done from the first heating cycle (Figure 1a) to obtain the thermal parameters of the starting material, just after 15 days from its processing, thus allowing secondary crystallization. In addition, the second heating cycle after a slow cooling (Figure 1b) allowed the thermal history of the material to be removed and the main thermal characterization parameters to be obtained.

Table 2 summarizes the main thermal parameters corresponding to the DSC thermograms of the first heating cycle plus an additional aging time of 15 days, obtained using DSC runs from Figure 1a. The thermogram of neat PCL showed a single endothermic peak at around 62 °C (*T*_m_PCL_), which was attributed to the melting of packed crystallites of PCL embedded in an amorphous PCL fraction. Since DSC tests were run from −50 °C, the *T*_g_PCL_ could not be clearly observed. This was because the *T*_g_PCL_ is very low, with values below −50 °C, so that, with this thermal program, it could not be accurately determined. On the other hand, the DSC thermogram of neat PHBH did not allow its *T*_g_PHBH_ to be identified either. This is because the crystalline fraction of PHBH remarkably increased after aging for 15 days; therefore, the remaining amorphous fraction was noticeably reduced, and then, the step in the baseline (around 0 °C), attributed to the *T*_g_PHBH_, could not be clearly seen. As one can see in Figure 1a, only the melt process of the crystalline fraction in PHBH could be observed with a peak located at 138 °C (*T*_m_PHBH_). Regarding binary PHBH/PCL blends, the two above-mentioned endothermal peaks could be seen in all the developed compositions: a first peak at around 60 °C, corresponding to melting of PCL, and a second peak at around 135 °C, related to the melting of PHBH. One can see that as the PCL wt % increased in the PHBH/PCL blends, the melt peak of PCL became larger, while inversely, the melt peak of PHBH was slightly diluted. These two independent peaks suggest some lack of miscibility between the two biopolyesters, as each polymer melted at its corresponding temperature [26,31].

The results in Table 2 indicate that neat PHBH is characterized by a small degree of crystallinity, *χ*_c_PHBH_ around 13%, even after the 15-day aging process at room temperature. According to the results of Xu et al. [27], this low *χ*_c_ was due to the irregularities in the structure of the polymer chain of the copolymer, which hinders the formation of crystallites, with increasing mol % 3-HH. The addition of PCL wt % slightly increases the crystallinity values, from 15% for the sample with 10 PCL wt % to 18% for 40 PCL wt %. Garcia-Garcia et al. [31] found similar results in the PHB/PCL system with an increase in the degree of crystallization *χ*_c_ from 55.1% to 58.2% with 25 wt % PCL. They attributed this to the fact that PCL can affect the crystallization kinetics of neat PHB. As expected, PHBH showed lower *χ*_c_ due to the hindering effects of 3-HH, as mentioned above. Contrary to this, Antunes et al. [39] reported a decrease in the degree of crystallinity of PHB by increasing PCL wt % up to 20 wt %, while an increase was observed for 30 wt % PCL. The thermograms in Figure 1a, are interesting as they clearly indicate the aging process after 15 days has been able to complete the secondary crystallization. Moreover, the results gathered in Table 2 corroborated the absence of secondary crystallization after 15 aging days.

With respect to the DSC thermograms obtained in a second heating cycle, shown in Figure 1b (after the cooling of the first heating cycle), it is worthy to note that these showed a clear change. In these DSC thermograms a step in the base line at about 0 °C could be clearly observed, which was attributable to the *T*_g___PHBH_ [38]. In this case, as the thermal history is completely different, the results regarding crystallinity were somewhat variable. Przybysz et al. [40] reported a remarkable decrease in PHB/PCL blends due to the addition of different peroxide-based compatibilizers, while Oyama et al. [26] showed completely different results for a PHBH/PCL system with peroxide-based compatibilizers, which showed an increase in *χ*_c_ of PHBH. Nevertheless, Antunes et al. [39] reported a decrease in *χ*_c_ of PHBH without any compatibilizer, which was attributed to changes in crystallization kinetics. In this work, we obtained somewhat varying effects of PCL wt % on the degree of crystallinity of PHBH, as its complex structure (hindering crystallization due 3-HH units) and the additional effects of PCL on crystallization kinetics could overlap with some simultaneous processes and lead to these changes. Obviously, PCL did not show its corresponding step change in the baseline as its *T*_g_PCL_ is lower than −50 °C. The compatibility of a polymer blend can be assessed by changes in *T*_g_ as Garcia et al. reported [41]. In this case, the addition of different PCL wt % to PHBH/PCL blends did not affect the *T*_g_PHBH_ values obtained, as shown in Table 3. 

The thermogram corresponding to 100% PHBH showed an exothermic peak at around 49 °C which stood for the peak temperature, *T*_cc_PHBH_, of the cold crystallization, with a crystallization enthalpy (Δ*H*_cc___PHBH_) of 21.14 J·g^−1^ [19]. At temperatures close to 113 °C, PHBH showed a first and small endothermic peak that corresponded to the melting of PHBH crystalline fraction (*T*_m1_PHBH_) and a second melt peak located at 120 °C (*T*_m2_PHBH_). These two endothermic peaks could be due to two effects. The first is based on the polymorphism presented by some copolymers such as PHBH as observed in other aliphatic polyesters. Due to the heterogeneous composition of the copolymer itself, different crystalline morphologies can be formed, with different thermal stability. The second effect is that crystallization produces primary crystals with low degree of perfection; these may melt and recrystallize to produce crystals of greater perfection or greater thickness, and this could be the explanation for the presence of two overlapped melting peaks [21,23,25,42].

On the other hand, the 100 PCL wt % sample only showed a very marked endothermic peak, at 55 °C, which corresponded to its melting temperature, *T*_m_PCL_, as mentioned above. Due to the similarity between the cold crystallization process of PHBH and the melting of the crystalline fraction of PCL, the binary PHBH/PCL blends showed two overlapped endothermal (PCL melting)-exothermal (PHBH cold crystallization) peaks in the DSC thermograms (Figure 1b). This overlapping did not allow either the melting enthalpy of the PCL (Δ*H*_m___PCL_) or the cold crystallization enthalpy of the PHBH (Δ*H*_cc___PHBH_) to be quantified accurately. This effect did not allow the correct calculation of *χ*_c_ in the PHBH/PCL system blends in this 2nd heating cycle as they were overlapped. Nevertheless, these 2nd heating DSC runs were interesting as the samples had undergone a thermal heating-cooling cycle to remove the thermal history, and subsequently, all the thermal transitions could be detected in a clearer way. In particular, the secondary crystallization of PHBH, which disappeared after the aging process (Figure 1a) and could not be detected, was clearly seen in the 2nd heating cycle.

On the other hand, the thermograms of the binary PHBH/PCL blends showed the characteristic melting peak corresponding to PCL and the two melting peaks of PHBH. It should be noted that as the PCL wt % in the PHBH/PCL blends increased, it had virtually no influence on the melting peak temperatures of PHBH (*T*_m1_PHBH_ and *T*_m2_PHBH_), which was the major component in the developed PHBH/PCL blends. The immiscibility between PHBH and PCL suggests that they form two separate phases with almost independent thermal parameters [30].

Regarding the enthalpies of the thermal transitions related to melting, the results obtained are shown in Table 3. The addition of small amounts of PCL (10 and 20 wt %) slightly increased the values of Δ*H*_m_PHBH_ *, which indicated a higher amount of energy to melt the crystalline fraction due to there was a slight increase in crystallinity. Higher amounts of PCL (30 and 40 wt %) offer the opposite effect leading to a decrease in crystallinity, probably due to changes in the crystallization kinetics due to the intrinsic structural complexity of PHBH (with 3-HH units which hinder crystallization) and PCL, which could affect crystallization, as has been described previously. When comparing these results with those obtained from samples aged for 15 days, the values of the melting enthalpies were slightly higher. Slow cooling favoured the phenomenon of cold crystallization of PHBH, so that the final melting enthalpy of the crystalline fraction was higher in samples with no previous thermal history.

The thermal stability of binary PHBH/PCL blends was analyzed using thermogravimetric analysis (TGA). Figure 2 presents a comparative plot of the TGA curves for neat PHBH and PCL and PHBH/PCL blends with different PCL wt %.

The TGA curve of neat PHBH shows a single-step thermal degradation process. The onset degradation temperature (*T*_2%_) for neat PHB was 264.6 °C. Once the thermal degradation/decomposition had started, it proceeded very fast with a maximum degradation rate temperature (*T*_deg_) of 301.3 °C. These results for individual PHBH were in accordance with those reported by Hosoda et al. [43] and Mahmood et al. [19]. Almost all the mass was thermally decomposed very quickly with an endset degradation temperature of 321 °C. A small residual char of 2.72 wt % was obtained for neat PHBH. It is worth mentioning that the thermal degradation of PHBH occurred in a very narrow temperature range of 57 °C. On the other hand, despite its crystalline fraction melting at relatively low temperature (58–60 °C), it is important to note that PCL was much more thermally stable since its onset degradation temperature (*T*_2%_) was 358 °C. In fact, PCL is one of the most thermally stable polyesters. Once the onset degradation temperature was reached, the thermal degradation proceeded in a single-step degradation process, with a maximum degradation rate of 430.9 °C (*T*_deg_), and also occurred in a very narrow range up to 474.8 °C, with a residual mass of 0.12 wt %.

The immiscibility between PHBH and PCL was reflected in the TGA curves of the blends. In all of them, two separated degradation steps could be observed: a first step at around 300–310 °C, which corresponded to the degradation of the PHBH phase, and a second step, at a higher temperature occurring in the 390–440 °C range, which corresponded to the degradation of PCL in the blend, as reported by Garcia-Garcia et al. [31] in P3HB-PCL binary blends. The immiscibility was clearly detected by observing the TGA curves. By observing the TGA curve of the 60PHBH-40PCL sample, the mass loss after the first degradation step was almost 40 wt %, which corresponded exactly to the PCL wt % in the blend. The same effect was observed in all other binary blends, which corroborated the lack of miscibility between PHBH and PCL. On the other hand, the blends also presented low residual char formation, between 2 and 1.13 wt %. Comparatively, there was a very slight shift of TGA thermograms to the right with increasing PCL wt % content.

The first derivative of the thermogravimetric curves (DTG), Figure 2b, shows two clearly differentiated thermal degradation processes without almost any overlapping, which corroborates the above-mentioned immiscibility of the PHBH/PCL binary blends. The first peak observed at temperatures between 301–307 °C corresponded to the maximum degradation rate temperature (*T*_deg_) of PHBH in all blends. On the other hand, the second peak, located between 420–430 °C, was attributed to the maximum degradation rate temperature of PCL in all the blends. There were very slight changes in the peak maximum values for each process, thus corroborating this immiscibility.

### 3.2. Thermomechanical Properties of PHBH/PCL Blends

In addition to the thermal stability, it is important to evaluate the effect of temperature on dimensional stability. To this end, samples of the different PHBH/PCL blends were subjected to a thermomechanical characterization, which allowed the coefficients of linear thermal expansion (CLTE) were obtained below and above the *T*_g_PHBH_ (Table 4).

As expected, at temperatures below *T*_g_PHBH_, the CLTE values were much lower than those obtained above *T*_g_PHBH_. The dimensional expansion of the material was lower below the characteristic *T*_g_, since the material offers a rigid, brittle, glassy behaviour. On the other hand, above *T*_g_, the polymer material changed its behaviour to a plastic, rubbery-like behaviour, and subsequently, the dimensional expansion was allowed. Neat PHBH showed a CLTE of 68 μm·m^−1^·°C^−1^ below its *T*_g_. As PCL is a much more flexible polymer, its addition to the PHBH/PCL blends provided increased flexibility, and subsequently, the CLTE increased according to PCL wt % contained in the blends. This increase in CLTE was proportional and increased up to 106.9 μm·m^−1^·°C^−1^ for the blend with 40 PCL wt %. This increase in the CLTE, which was directly proportional to the PCL wt %, was representative for a somewhat loss of fragility and an improvement in the ductile behaviour of PHBH at low temperatures. The same effect was observed for the CLTE values obtained at temperatures above *T*_g_PHBH_ in the blends. In this case, neat PHBH offered a remarkable increase in its CLTE to 172.5 μm·m^−1^·°C^−1^, which was remarkably higher than the CLTE below its *T*_g_ (68 μm·m^−1^·°C^−1^). The increasing tendency for the CLTE was similar to that mentioned above, and the PHBH/PCL blend with 40 PCL wt % showed the maximum CLTE of about 198.9 μm·m^−1^·°C^−1^.

The thermal dynamic mechanical analysis of the PHBH/PCL blends allowed the variation of the storage modulus, *E’*, and the damping factor (tan δ) to be obtained. The damping factor was directly related to the phase angle (δ), which is representative for the delay between the applied dynamic stress (σ_d_) and the obtained dynamic elongation (ε_d_). The DMTA technique is much more sensitive to the glass transition temperature, *T*_g_, detection since this technique measures changes in mechanical properties as a function of temperature, which includes the definition of *T*_g_ with a change from a glassy state to a rubber-like behaviour [30,31,34,37,38]. The dependence of *E’* on temperature is shown graphically in Figure 3a.

The values of *E’* obtained for neat PHBH at very low temperatures (e.g., −70 °C) were high, about 2 GPa, since this temperature zone corresponds to an elastic-glassy behaviour of the material, which means it was far (below) from its *T*_g_. For example, at −40 °C, the *E’* of neat PHBH was 1903.4 MPa. For this same temperature, the *E’* for neat PCL was much lower, with a value of 596.3 MPa. In this case, the material showed a visco-elastic behaviour, due to it was above *T*_g_ (*T*_g_PCL_ about −60 °C). Thus, adding different PCL wt % to the PHBH led to blends with a clear decreasing tendency of *E’* which was proportional to the PCL wt %. As the temperature increased, a remarkable decrease in *E’* values was observed, which represented the change from a glassy state (rigid with high *E’* values) to a viscous, rubber-like behaviour (viscoplastic with low *E’* values). Obviously, this was directly related to the glass transition temperature range from −10 to 20 °C. By taking the *T*_g_ criterium corresponding to the peak maximum of the dynamic damping factor, *T*_g_PHBH_ was 9 °C. Since the *T*_g_ of both PHBH and PCL is relatively low, at room temperature both polymers are in the rubbery-plateau zone, with a relatively flexible viscoelastic behaviour, as shown in the respective *E’* plots. This is the typical behaviour of polymers above their *T*_g_, as Burgos et al. and Avolio et al. reported [44,45]. At 25 °C, the value of *E’* was 653.2 MPa, almost three times lower than neat PHBH below its *T*_g_. The same ranges of variation were maintained for the studied blends, with values between [539 MPa, 398 MPa] at 25 °C vs. [1666 MPa, 1289 MPa] at −40 °C. At higher temperatures, *E’* tended to have very low values (close to 0 MPa) for blends with high PCL wt %. This effect was because PCL melted at about 60 °C, and once this temperature was overpassed, PCL changed from a rubber-like state to a melt state with extremely low elastic properties. Nevertheless, in blends with 10 and 20 wt % PCL (it is important to bear in mind that PHBH content in these blends was still very high and is not highly affected by PCL addition in terms of dynamic-mechanical behaviour at high temperatures) the effect was less pronounced so that at 60 °C, *E’* was 178.5 and 142.0 MPa respectively. These values are typical of a rubber-like material and decrease as the wt % PCL increases.

Figure 3b shows the variation of the dynamic damping factor (tan δ) as a function of temperature for neat PHBH and PCL and for their blends with different PCL wt %. With respect to the neat PCL curve, a marked and broad peak was observed with a peak maximum located at −47 °C, corresponding to the glass transition temperature *T*_g_ of PCL [30,31] With regard to neat PHBH, a narrow peak (compared to that of PCL) can be seen, with its maximum peak value located at 9 °C. As mentioned above, all the applied techniques suggested poor (or even lack) of miscibility between these two polymers. Therefore, the changes in the respective *T*_g_ values of neat PCL (−47 °C) and neat PHBH (9 °C) were not changed in a significant way, therefore corroborating the lack of miscibility of PHBH/PCL blends [38]. In addition, close to 60 °C, PHBH/PCL blends show a third peak which was related to the partial melting of one of the components, i.e., PCL. This third peak was much more pronounced in blends with high PCL wt % and shows proportionality to the PCL wt %, but this third peak appears always at the same temperature of about 60 °C.

### 3.3. Mechanical Properties and Morphology of PHBH/PCL Blends

Table 5 presents a summary of the mechanical properties obtained from tensile, flexural, and hardness (Shore D) tests, corresponding to neat PHBH and PCL and PHBH/PCL binary blends with different PCL wt %. All the blends of this binary system offered a noticeable decrease in the tensile modulus (*E*_t_) with an increase in the PCL wt % content compared to neat PHBH. PHBH/PCL blends with 40 PCL wt % showed an *E*_t_ value of 722 MPa, which was remarkably lower than that of neat PHBH at about 1022 MPa. This represented a % decrease of almost 30%. The addition of PCL resulted in lowering the overall stiffness of the PHBH/PCL blends. Similar results were reported by Hinüber et al. in PHBH/PCL blends [46]. It should be noted that neat PCL is a biopolymer characterized by a very low tensile strength (σ_t_) of 12.2 MPa (typical of a rubber-like polymer), high elongation at break (ε_b_) (no break means more than 600% as this is the maximum elongation in the used machine), and a low tensile modulus (*E*_t_) of 386 MPa. All these properties positively contributed to improving the ductility of neat PHBH by blending with PCL. The same tendency could be observed for the tensile strength of the PHBH/PCL blends. The addition of different PCL wt % progressively decreased the values of σ_t_, from 16 MPa for neat PHBH to [13.4, 13.9] MPa with 20 PCL wt % and 40 PCL wt %, respectively. This decrease in the resistance parameters was due to the two-phase structure of the blends, resulting from the lack of (or very poor) miscibility, as mentioned above. The immiscibility between PHBH and PCL forms a dispersed PCL phase that interrupts the continuity of the PHBH-rich matrix phase, making the stress transfer difficult and, subsequently, decreasing the *E*_t_ and σ_t_ of the PHBH/PCL blends with increasing PCL wt %.

With respect to the elongation at break, ε_b_(%), the effect was opposite and very positive. The increase in ε_b_(%) in the PHBH/PCL blends also increased ductility with PCL wt %. PHBH is a rather fragile polymer (after aging or secondary crystallization) with only 13.9% elongation at break. Some researchers attributed this fragility to the secondary crystallization or aging on PHBH, which reduced the amorphous fraction [18,19]. With the addition of only 20 PCL wt %, the ε_b_(%) increased up to 67.8% (which represented a percentage increase of almost 387%). Even more, the ε_b_(%) for the blend with the highest PCL wt % considered in this study, i.e., 40 wt %, showed an ε_b_(%) of 461% (this was 33 times higher than neat PHBH).

Table 5 also offers the flexural parameters of neat PHBH and PCL and the PHBH/PCL blends. As in tensile conditions, PHBH/PCL blends became less rigid with PCL addition. This can be confirmed by a clear decrease in the flexural modulus (*E*_f_). Neat PHBH had an *E*_f_ value of 1029 MPa, and this decreased progressively to 801.7 MPa for the PHBH/PCL blend containing 40 PCL wt %. In this case, the decrease in the flexural strength (σ_f_) was not so pronounced as that observed in tensile conditions. 

In addition, the Shore D hardness, as it is a mechanical resistant property, it followed the same tendency as that observed for both modulus and strength. PHBH showed a Shore D of 61, and the addition of the flexible PCL to PHBH decreased the Shore D values down to 55 for the blend with 40 PCL wt %. Despite this, all Shore D values were close to 58 with very slight variations.

Another important property of polymers is toughness. Figure 4 shows the variation of the absorbed energy per unit area (impact resistance) obtained using a Charpy’s test. PHBH is rather brittle, and consequently, it offers low toughness. Graphically, an interesting increase in the impact resistance of PHBH/PCL blends could be observed as the PCL wt % increased. It is important to bear in mind that the energy absorption capacity under impact conditions is directly related to the plastic deformation capacity of the material before the breakage occurs, and the supported stress, too [47]. The presence of a biphasic structure (as reported in morphology analysis) could contribute to improving the toughness, as Ferri et al. reported [48]. Thus, the results obtained corroborated those previously analyzed for tensile and flexural characterizations. Neat PHBH showed a low impact resistance of 5.56 kJ·m^−2^; with the addition of only 10 PCL wt %, it was increased to 8.7 kJ·m^−2^ (which represented a % increase of 56%). This same trend was proportionally maintained as the PCL wt % increased. It is worth noting that the impact energy for the PHBH/PCL blend with 40 PCL wt % was around 10.5 kJ·m^−2^, almost twice as much as neat PHBH. These results were consistent with the above-mentioned decrease in the intrinsic brittleness of neat PHBH by blending with PCL [37].

Figure 5 shows the FESEM images of the impact fracture surfaces of the PHBH/PCL binary blends with different PCL wt %. All the images show a structure with a continuous and homogeneous matrix phase, which corresponded to the PHBH, and a scattered phase of special morphology, which corresponded to the PCL. This two-phase structure confirmed the lack of miscibility between PHBH and PCL as already concluded with previous thermal analyses. Similar findings were proposed by Quiles-Carrillo et al. [49]. Nevertheless, the typical drop-like structure could not be observed in this system. In addition, the special morphology of the dispersed phase of the PCL, forming small, thin “sheets or flakes” homogeneously distributed with very regular sizes, must be emphasized. FESEM images also revealed that the higher the PCL wt %, the greater the amount of dispersed phase that could be observed.

To check that this dispersed phase corresponded to PCL present in the binary blends, a selective PCL extraction with acetone was performed for 24 h [38]. Figure 6 shows the FESEM images obtained after this selective extraction. In all of them, the dispersed phase with small flake-like shapes was no longer observed. Instead, small and thin empty voids appeared, which corresponded exactly to the geometric shape of the PCL phase before the selective attack (Figure 5). This observation allowed us to conclude that the dispersed phase indeed corresponded to the PCL present in the binary blend. The lack of miscibility between the biopolymers of the PHBH/PCL system was responsible for the internal biphasic structure formed in the blends. In addition, FESEM images showed how the dispersed PCL-rich phase interrupted the continuity of the PHBH matrix, so that the stress transmission inside the material when subjected to external stresses was not adequate [38]. This reduced the mechanical resistant parameters, which corroborated the results obtained in the mechanical characterization of the PBHB/PCL binary blends.

## 4. Conclusions

The processing and obtaining of PHBH/PCL binary blends allowed a new environmentally friendly material to be obtained with improved toughness and ductile properties, suitable for industrial use in the packaging sector and for medical applications too.

Considering the intrinsic fragility of PHBH, the addition of different PCL wt %, allowed its toughness, ductility, and, above all, impact resistance to be improved. Blends with a 10 PCL wt % offered a percentage increase in the impact resistance of about 56%. The impact resistance was even improved up to double the initial value of neat PHBH by adding 40 PCL wt %. This effect of increasing the ductility of PHBH by increasing the PCL content in the PHBH/PCL blends had an opposite effect on mechanical resistant properties such as modulus and strength (tensile and flexural). In contrast, the elongation at break was remarkably improved, from 13.9% for neat PHBH up to 461% for the PHBH/PCL blend with 40 PCL wt %.

On the other hand, thermal analysis suggested high immiscibility between PHBH and PCL, since the main thermal parameters of neat PHBH and PCL remained unchanged in blends, which is characteristic of very poor or lack of miscibility. Only a slight increase in the thermal stability of neat PHBH was obtained by adding different PCL wt %, since PCL is much more thermally stable than most biopolyesters. The dynamic mechanical thermal analysis (DMTA) allowed accurate values of *T*_g_ to be obtained, with two clear and unchanged values located at −47 °C (*T*_g_ of PCL) and at 9 °C (*T*_g_ of PHBH). These characteristic *T*_g_ values remained unchanged in the PHBH/PCL blends, thus suggesting lack of miscibility. Regarding the morphology of the PHBH/PCL blends, they did not show the typical drop-like PCL phase embedded in a PHBH matrix. PCL appeared in the form of small flakes which could exert a positive effect on ductile properties.

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
