# Peer review of "Manufacturing and Properties of Binary Blend from Bacterial Polyester Poly(3-hydroxybutyrate-co-3-hydroxyhexanoate) and Poly(caprolactone) with Improved Toughness"

_polymers, 2020, doi:10.3390/polym12051118_

Round 1

Reviewer 1 Report

Authors report here the preparation of polymer blends based on poly(3-hydroxybutyrate-co-3-hydroxyhexanoate) (PHBH) and poly(ε-caprolactone) (PCL). PHBH is one of the less studied poly(hydroxialcanoates) (PHAs) and thus the main novelty of this research is focused on improving it performance to extend PHBH in industrial applications. The PHBH-PCL were fully characterized by different techniques, including thermal, mechanical, thermo-mechanical and structural characterization. The work is well planed and organized which make it easy to reading. The obtained results are very interesting for packaging  applications were a high flexibility is required. The literature cited is updated and related with the field of application. In my opinion the manuscript is of high quality and deserve of publication in Polymers in present form.

I only have some formal comments:

Line 16: in "Poly (3-hydroxybutyrate-co-3-hydroxyhexanoate)" the space between Poly and (3-hydroxybutyrate-co-3-hydroxyhexanoate)should be deleted.

Line 301: "%wt" should be "wt%" to be the same in the whole manuscript.

Author Response

May11, 2020

Dear colleagues,

            Thank you very much for your e-mail dated on May 9, 2020 regarding the manuscript entitled “Manufacturing and properties of binary blend from bacterial polyester poly(3-hydroxybutyrate-co-3-hydroxyhexanoate) and poly(caprolactone) with improved toughness which was allocated with reference number polymers-804095.

            We are sending the revised version of the manuscript that includes all suggestions and corrections proposed by the reviewers. In addition, an in depth check of the grammar and spelling has been carried out and all detected mistakes have been corrected. Changes done to manuscript have been emphasized in yellow in order to facilitate their searching. We have worked in accordance with all reviewer comments, so we consider that the version we are sending to you includes all necessary changes.

Sincerely,

PhD Student Juan Ivorra-Martinez

Institute of Materials Technology

Universitat Politècnica de València

Plaza Ferrándiz y Carbonell 1, 03801, Alcoy, Alicante (Spain)

e-mail: juaivar@doctor.upv.es

Reviewer 1

Authors report here the preparation of polymer blends based on poly(3-hydroxybutyrate-co-3-hydroxyhexanoate) (PHBH) and poly(ε-caprolactone) (PCL). PHBH is one of the less studied poly(hydroxialcanoates) (PHAs) and thus the main novelty of this research is focused on improving it performance to extend PHBH in industrial applications. The PHBH-PCL were fully characterized by different techniques, including thermal, mechanical, thermo-mechanical and structural characterization. The work is well planed and organized which make it easy to reading. The obtained results are very interesting for packaging  applications were a high flexibility is required. The literature cited is updated and related with the field of application. In my opinion the manuscript is of high quality and deserve of publication in Polymers in present form.

I only have some formal comments:

Line 16: in "Poly (3-hydroxybutyrate-co-3-hydroxyhexanoate)" the space between Poly and (3-hydroxybutyrate-co-3-hydroxyhexanoate)should be deleted.

Line 301: "%wt" should be "wt%" to be the same in the whole manuscript.

ANSWER

We thank the reviewer for detecting these typos. Accordingly, the above-mentioned typos have been corrected as well as other mistakes mentioned by other reviewers. Moreover, we have done a double-check to detect additional grammar mistakes and/or typos, and the version we are sending has improved with this regard.

Reviewer 2 Report

The paper “Manufacturing and properties of binary blend from bacterial polyester poly(3-hydroxybutyrate-co-3-hydroxyhexanoate) and poly(caprolactone) with improved toughness” by J. I. Martinez presents a thermal and mechanical properties of PHBH/PCL blends with different compositions were studied elaborated by extrusion followed by injection moulding. The manuscrit is clear and well written, all the conclusions are supported by the data and can be published after minor revisions listed below:

Minor comments:

  • Change line with dot in Figure 4.
  • The melting enthalpies are counted negative. I understand that this is purely conventional but generally endothermal events are counted as positive.

Author Response

May 11, 2020

Dear colleagues,

            Thank you very much for your e-mail dated on May 9, 2020 regarding the manuscript entitled “Manufacturing and properties of binary blend from bacterial polyester poly(3-hydroxybutyrate-co-3-hydroxyhexanoate) and poly(caprolactone) with improved toughness which was allocated with reference number polymers-804095.

            We are sending the revised version of the manuscript that includes all suggestions and corrections proposed by the reviewers. In addition, an in depth checks of the grammar and spelling has been carried out and all detected mistakes have been corrected. Changes done to manuscript have been emphasized in yellow in order to facilitate their searching. We have worked in accordance with all reviewer comments so we consider that the version we are sending to you includes all necessary changes.

Sincerely,

PhD Student Juan Ivorra-Martinez

Institute of Materials Technology

Universitat Politècnica de València

Plaza Ferrándiz y Carbonell 1, 03801, Alcoy, Alicante (Spain)

e-mail: juaivar@doctor.upv.es

Reviewer 2

The paper “Manufacturing and properties of binary blend from bacterial polyester poly(3-hydroxybutyrate-co-3-hydroxyhexanoate) and poly(caprolactone) with improved toughness” by J. I. Martinez presents a thermal and mechanical properties of PHBH/PCL blends with different compositions were studied elaborated by extrusion followed by injection moulding. The manuscrit is clear and well written, all the conclusions are supported by the data and can be published after minor revisions listed below:

Minor comments:

Change line with dot in Figure 4.

Answer

We thank the reviewer for this comment. As indicated by the reviewer, the corresponding points to construct the plot evolution of the absorbed-energy values as a function of increasing PCL wt%, have been clearly identified by dots. The line has been changed to a dashed line while the error bars have been left as they are to show the standard deviation from the average point.

The melting enthalpies are counted negative. I understand that this is purely conventional but generally endothermal events are counted as positive.

Answer

We totally agree with the reviewer regarding this comment. It is evident that the sign is directly related to the type of process (endotherm and exotherm offer opposite signs). So that, as the melt enthalpies are clearly endotherm, it is not necessary to count them as negative values. Therefore, as recommended by the reviewer, the melting enthalpies have been counted as positive on the different tables, in the revised version we are sending.

Reviewer 3 Report

In this work, a flexible petroleum-derived polyester, namely poly(caprolactone) is used to obtain PHBH/PCL blends with different compositions (from 0 to 40 PCL wt%) by extrusion followed by injection moulding. The thermal analysis of the binary blends was studied by means of DSC and TGA. DSC reveled immiscibility between PCL and PHBH. The mechanical-dynamic thermal analysis (DMTA) allowed a precise determination of the glass transition temperatures (Tg) as a function of the blend composition. By means of FESEM, an internal structure formed by two phases has was observed, with a matrix phase of PHBH and a finely dispersed PCL phase. These results confirm the immiscibility between these two biopolymers. As expected, an increase in ductile properties, provides decreased some mechanical resistant properties, e.g. the tensile modulus and strength, the flexural modulus and strength decrease with the increasing of PCL wt% in PHBH/PCL blends.

  1. Different amounts (wt%) of PHBH and PCL. Why is the ratio 50PHBH-50PCL not set?
  2. “from 30 °C up to 700 °C”. As can be seen from Figure 2, this should be 600 °C instead of 700 °C.
  3. As can be seen from Table 3, There is no obvious change in Hm_PHBH*(°C) values of 90PHBH-10PCL and 80PHBH-20PCL. The authors should explain this result.
  4. Table 3 in Page 10. The author should standardize the writing of numbers for CLTE below Tg_PHBH.
  5. “This effect is very marked, due to the visco-elastic behavior of the blends.”. Some references are needed.
  6. “For PHBH/PCL blends with low PCL wt%, this effect is delayed, so that around 100 °C the E' values are close to 90 MPa.”. The authors should explain this result.
  7. The authors reported a lot of data in “3. Results” sections. However, discussion should also be addressed to compare with the studies in the literatures.
  8. Many irregularities were described in the manuscript. For example, “These results confirm” should be “These results confirmed”, “of 2.72 wt% is obtained” should be “of 2.72 wt% was obtained”, and so on.

Author Response

May 11, 2020

Dear colleagues,

            Thank you very much for your e-mail dated on May 9, 2020 regarding the manuscript entitled “Manufacturing and properties of binary blend from bacterial polyester poly(3-hydroxybutyrate-co-3-hydroxyhexanoate) and poly(caprolactone) with improved toughness which was allocated with reference number polymers-804095.

            We are sending the revised version of the manuscript that includes all suggestions and corrections proposed by the reviewers. In addition, an in depth checks of the grammar and spelling has been carried out and all detected mistakes have been corrected. Changes done to manuscript have been emphasized in yellow in order to facilitate their searching. We have worked in accordance with all reviewer comments so we consider that the version we are sending to you includes all necessary changes.

Sincerely,

PhD Student Juan Ivorra-Martinez

Institute of Materials Technology

Universitat Politècnica de València

Plaza Ferrándiz y Carbonell 1, 03801, Alcoy, Alicante (Spain)

e-mail: juaivar@doctor.upv.es

Reviewer 3

In this work, a flexible petroleum-derived polyester, namely poly(caprolactone) is used to obtain PHBH/PCL blends with different compositions (from 0 to 40 PCL wt%) by extrusion followed by injection moulding. The thermal analysis of the binary blends was studied by means of DSC and TGA. DSC reveled immiscibility between PCL and PHBH. The mechanical-dynamic thermal analysis (DMTA) allowed a precise determination of the glass transition temperatures (Tg) as a function of the blend composition. By means of FESEM, an internal structure formed by two phases has was observed, with a matrix phase of PHBH and a finely dispersed PCL phase. These results confirm the immiscibility between these two biopolymers. As expected, an increase in ductile properties, provides decreased some mechanical resistant properties, e.g. the tensile modulus and strength, the flexural modulus and strength decrease with the increasing of PCL wt% in PHBH/PCL blends.

1 Different amounts (wt%) of PHBH and PCL. Why is the ratio 50PHBH-50PCL not set?

ANSWER

We thank the reviewer for this comment. Regarding to this, PCL is just used to improve toughness and other ductile properties; therefore we selected a maximum wt% addition of PCL of 40 wt% since the 50/50 could represent the phase transition in which, PCL could govern the overall properties of the developed binary blends. That is why authors selected up to 40 wt% PCL. Our intention was not to cover the whole PHBH/PCL system which could be interesting for further researches. The main aim was to improve the low toughness of neat PHBH after the secondary crystallization.

According to this and, with the aim of clarifying this issue, the following paragraph has been included in the revised version

“As it has been reported in other works, the typical weight content (wt%) of flexible polymer blended with brittle polymers to improve toughness is comprised between the of 20-40 wt% range. In this work we selected a maximum PCL content of 40 wt% since at this composition, PCL is still the minor component in the blend [32-34]. Garcia et al. [31] studied the whole PHB/PCL system and they revealed, as expected, that above 50 wt% PCL, it is PCL which defines the properties of the blend. Ferry et al. [35] also confirmed a maximum loading of 30 wt% PCL to improve the high brittleness of neat PLA.

2 “from 30 °C up to 700 °C”. As can be seen from Figure 2, this should be 600 °C instead of 700 °C.

ANSWER

We thank the reviewer for this comment. Initially, we run these TGA test from 30 ºC up to 700 ºC at a constant heating rate of 20 ºC/min. Nevertheless, the above 550 ºC no relevant change was observed; therefore, we decided to plot the region up to 600 ºC since the TGA information above 600 ºC did not provide any relevant information. Nevertheless, as recommended by the reviewer, the TGA plots and the DTG plots have been plotted again using the whole scale, as indicated in the experimental section, i.e., from 30 ºC to 700 ºC.

3 As can be seen from Table 3, There is no obvious change in Hm_PHBH*(°C) values of 90PHBH-10PCL and 80PHBH-20PCL. The authors should explain this result.

ANSWER

We are in total agreement with this comment. Firstly, we have gone back to the raw data to detect any mistake, but the obtained data is correct. Secondly, we have searched for related literature with similar systems, PHBH/PCL and PHB/PCL (compatibilized and uncompatibilized) and surprisingly we have found that there is not a clear effect of PCL on the degree of crystallinity of PHBH. We have found literature with opposite results regarding this issue and the explanation they provide is that PCL can affect the crystallinity kinetics. Therefore, we have included three different paragraphs showing this controversy. In fact, this is an interesting issue to work with in further studies since the degree of crystallinity can have an important effect on other properties.

“The results in Table 2 indicate that neat PHBH is characterized by a small degree of crystallinity, cc_PHBH around 13%, even after the 15-day aging process at room temperature. According to the results of Xu et al. [27], this low cc was due to the irregularities in the structure of the polymer chain of the copolymer, which hinders the formation of crystallites, with increasing mol% 3-HH. The addition of PCL wt% slightly increases the crystallinity values, from 15% for the sample with 10 PCL wt% to 18% for 40 PCL wt%. Garcia-Garcia et al. [31] found similar results in the PHB/PCL system with an increase in the degree of crystallization cc from 55.1% to 58.2% with 25 wt% PCL. They attributed this to the fact that PCL can affect the crystallization kinetics of neat PHB. As expected, PHBH showed lower cc due to the hindering effects of 3-HH as above-mentioned. Contrary to this, Antunes et al. [39] reported a decrease in the degree of crystallinity of PHB by increasing PCL wt% up to 20 wt%, while an increase was observed for 30 wt% PCL. The thermograms in Figure 1a, are interesting as they clearly indicate the aging process after 15 days, has been able to complete the secondary crystallization. Moreover, the results gathered in Table 2 corroborated the absence of secondary crystallization after 15 aging days.

With respect to the DSC thermograms obtained in a second heating cycle, Figure 1b (which corresponds to the same heating as Figure 1a plus a cooling stage from 200 °C to -50 °C, and then the 2nd heating), it is worthy to note that these showed a clear change. In these DSC thermograms a step in the base line at about 0 °C could be clearly observed, which was attributable to the Tg-PHBH [38]. In this, case, as the thermal history is completely different, the results regarding crystallinity were somewhat variable. Przybysz et al. [40] reported a remarkable decrease in PHB/PCL blends due to the addition of different peroxide-based compatibilizers, while Oyama et al. [26] showed completely different results on PHBH/PCL system with peroxide-based compatibilizers, which showed an increase in cc of PHBH. Nevertheless Antunes et al. [39] reported a decrease in cc of PHBH without any compatibilizer, which was attributed to changes in crystallization kinetics. In this work, we obtained somewhat varying effects of PCL wt% on the degree of crystallinity of PHBH as its complex structure (hindering crystallization due 3-HH units) and the additional effects of PCL on crystallization kinetics, could overlap some simultaneous processes and lead to these changes. Obviously, PCL did not show its corresponding step change in the baseline as its Tg_PCL is lower than -50 °C. The compatibility of a polymer blend can be assessed by changes in Tg as Garcia et al. reported [41]. In this case, the addition of different PCL wt% to PHBH/PCL blends did not affect the Tg_PHBH values obtained, as shown in Table 3.

Regarding the enthalpies of the thermal transitions related to melting, the results obtained are shown in Table 3. The addition of small amounts of PCL (10 and 20 wt%), slightly increased the values of DHm_PHBH*, which indicated the need of more energy to melt the crystalline fraction or, what is the same, there was a slight increase in crystallinity. Higher amounts of PCL (30 and 40 wt%), offers the opposite effect by a decrease in crystallinity, probably due to changes in the crystallization kinetics due to the intrinsic structural complexity of PHBH (with 3-HH units which hinder crystallization), and PCL which could affect crystallization as it has been described previously.”

 4 Table 3 in Page 10. The author should standardize the writing of numbers for CLTE below Tg_PHBH.

ANSWER

We thank the reviewer for realizing about this. In accordance, we have changed the “, “ by “.”

5 “This effect is very marked, due to the visco-elastic behavior of the blends.”. Some references are needed.

ANSWER

We totally agree with this suggestion.  Accordingly, the sentence has been rewritten with some new supporting literature. With this, the actual behaviour os a polymer above its Tg is clearer and this can avoid some misunderstanding.

“Since the Tg of both PHBH and PCL is relatively low, at room temperature both polymers are in the rubbery-plateau zone, with a relatively flexible visoelastic behavior, as shown in the respective E' plots. This is the typical behaviour of polymers above their Tg. as Burgos et al. and Avolio et al. reported [44,45]”

6 “For PHBH/PCL blends with low PCL wt%, this effect is delayed, so that around 100 °C the E' values are close to 90 MPa.”. The authors should explain this result.

ANSWER

We agree with the reviewer. To a better understanding we have rewritten the corresponding paragraph to avoid misunderstanding.

“Nevertheless, in blends with 10 and 20 wt% PCL (it is important to bear in mind that PHBH content in these blends is still very high and is not highly affected by PCL addition in terms of dynamic-mechanical behaviour at high temperatures) the effect was less pronounced so that at 60 °C E’ was 178.5 MPa and 142.0 MPa respectively. These values are typical of a rubber-like material and decrease as the wt% PCL increases.”

7 The authors reported a lot of data in “3. Results” sections. However, discussion should also be addressed to compare with the studies in the literatures.

ANSWER

We agree with this suggestion. As in many other journals, Results & discussion sections are commented at the same time. Therefore, we have renamed section “3. Results” to section “3. Results & discussion”.

8 Many irregularities were described in the manuscript. For example, “These results confirm” should be “These results confirmed”, “of 2.72 wt% is obtained” should be “of 2.72 wt% was obtained”, and so on.

ANSWER

We are in total agreement with the reviewer; there were too many grammatical mistakes, unfinished sentences, and so on. Sometimes it was difficult to understand some sentences. Therefore, as recommended by the reviewer we have carefully read (double-checked) the manuscript to avoid these type of grammar mistakes, paying special attention to tenses, verbs, and coherence between them. All changes are yellow-marked in the revised version.

Round 2

Reviewer 3 Report

Based on the author's modification, this article can be accepted.